# A Smartphone-Based Disposable Hemoglobin Sensor Based on Colorimetric Analysis

**DOI:** 10.3390/s23010394

**Published:** 2022-12-30

**Authors:** Zhuolun Meng, Muhammad Tayyab, Zhongtian Lin, Hassan Raji, Mehdi Javanmard

**Affiliations:** Department of Electrical and Computer Engineering, Rutgers University, Piscataway, NJ 08854, USA

**Keywords:** biosensor, hemoglobin, lab-on-a-chip, medical tape, micropump, rapid test, smartphone colorimetric measurement

## Abstract

Hemoglobin is a biomarker of interest for the diagnosis and prognosis of various diseases such as anemia, sickle cell disease, and thalassemia. In this paper, we present a disposable device that has the potential of being used in a setting for accurately quantifying hemoglobin levels in whole blood based on colorimetric analysis using a smartphone camera. Our biosensor employs a disposable microfluidic chip which is made using medical-grade tapes and filter paper on a glass slide in conjunction with a custom-made PolyDimethylSiloaxane (PDMS) micropump for enhancing capillary flow. Once the blood flows through the device, the glass slide is imaged using a smartphone equipped with a custom 3D printed attachment. The attachment has a Light Emitting Diode (LED) that functions as an independent light source to reduce the noise caused by background illumination and external light sources. We then use the RGB values obtained from the image to quantify the hemoglobin levels. We demonstrated the capability of our device for quantifying hemoglobin in Bovine Hemoglobin Powder, Frozen Beef Blood, and human blood. We present a logarithmic model that specifies the relationship between the Red channel of the RGB values and Hemoglobin concentration.

## 1. Introduction

Hemoglobin is one of the most vital components in blood that carries oxygen from the lungs to the rest of the human body. Therefore, an imbalance in hemoglobin levels can lead to severe health problems [1]. Normal ranges for hemoglobin vary but are generally between 13.8 and 17.2 grams per deciliter (g/dL) for males and 12.1 to 15.1 g/dL for females [2]. It is an important biomarker for various diseases such as anemia [3], sickle cell disease [4], and thalassemia [1]. Hemoglobin tests are usually conducted in a laboratory setting. These tests are conducted using expensive equipment such as Automated Hematology Analyzers (AHAs) which can cost anywhere between $2000–$15,000 [5]. Moreover, such equipment requires highly trained professionals to operate and interpret results. Therefore, there is a need to perform hemoglobin tests regularly with ease in a Point-of-Care setting. Advancements in healthcare technology have enabled the production and commercialization of portable hemoglobin analyzers [6]. Although such types of analyzers are a good option for rapid assessment by people with hemoglobin-related diseases, these are generally expensive and difficult to operate presenting a barrier to use by the general public.

In the past few decades, technology has evolved to meet the requirements and challenges of building Lab-on-a-Chip (LOC) and microfluidic devices. These devices and biosensors have been developing rapidly to manipulate small amounts of fluids for analytical chemistry and medical detection [7,8,9,10,11,12]. Specifically, microfabrication techniques have a huge impact on the types of biosensors and devices being introduced for biological assays [13,14,15]. Microfabrication techniques generally require a cleanroom environment to operate. Although robust and efficient, the cleanroom environment is very difficult and costly to maintain. Furthermore, using a cleanroom requires expensive materials and adept operators. The soft microfluidic channels which are typically composed of PDMS are explored as an alternative to silicon fabrication because elastomers are inexpensive, transparent, inert, biocompatible, and easy and fast to fabricate. Baking or oxide plasma is required to assemble PDMS channels onto other substrates. However, one of the widely known problems with using PDMS is the bonding issues encountered when bonding the PDMS to other substrates [16,17,18,19,20,21]. Besides, PDMS requires a lot of time to turn to a solid-state. Moreover, the quality is hard to control because of the bubbles formed during the curing process and it is generally impossible to get rid of these bubbles completely. Additionally, molds are used to create expected features in PDMS, and in order to change the shape, a new mold needs to be created.

An alternative approach is to use devices that are made of polymer materials and paper [22,23,24]. These devices have highly desirable characteristics such as being disposable, easy to fabricate, and having a low risk of mutual contamination making them ideal to be used in a Point-of-Care setting [25,26,27,28]. Researchers have explored the use of smartphone cameras to measure hemoglobin levels from blood or the skin [29,30,31]. However, these approaches do not have control over the external light sources and background illumination, which will heavily affect the results.

In this paper, we describe a hemoglobin concentration diagnostic device using colorimetric measurement; it is completely user-friendly without needing any expensive and bulky equipment for detection—only a smartphone camera. Our approach to quantifying hemoglobin levels addresses the challenges discussed above and satisfies the rules of ASSURED criteria (Affordable, Sensitive, Specific, User-friendly, Rapid and robust, Equipment-free, and Deliverable) by the WHO [32]. The design and fabrication process is simple and easy. We use SolidWorks to design the features and a laser cutter is used for cutting out the features on medical tapes. Furthermore, the shapes can be changed readily and in a facile manner. Compared with cleanroom microfabrication and PDMS, fabrication using the medical tapes is also economical, the price of which can be lowered to 0.24 USD per piece. To accelerate the reaction in the channel without the need for a redundant solution, we used PDMS micropumps to control and direct the solution [33]. Furthermore, the 3D-printed portable phone clamp can eliminate other lighting noises [34,35,36]. Using ImageJ, we obtain red-channel values from the images and obtain a logarithmic regression model that relates the hemoglobin concentration to the red-channel values.

## 2. Principle

In this section, the principles for the whole system are described with the use of conceptual pictures including the diagnostic devices, optical instruments, and colorimetric measurement.

### 2.1. Device Principle

The conceptual figure of the device is shown in Figure 1a. The microfluidic channels are formed by a laser cutter and then attached to the glass slides, which are embedded with filter paper (Fisherbrand Filter Paper P8, Fisher Scientific International, Inc., Pittsburgh, PA, USA) inside the channels. To operate and suck the solution into the channels, a PDMS micropump and PDMS extension are mounted onto the device. In Figure 1b, the glass capillary is shown to be approaching the fingers to suck the blood into the sensor. In Figure 1c, the thumb is shown to be pressing the micropump to vent the air when the glass capillary is put into the blood to create an environment with low atmospheric pressure. Then, the thumb is released to vacuum the blood into the channel to immerse the filter paper, and put the chip into the 3D printed phone clamp as shown in Figure 1d. After reacting, the whole device is slipped into the phone clamp to continue the colorimetric measurement process.

### 2.2. Optical Instrument

The purpose of using an optical instrument is to eliminate external light noise and control the light environment’s consistency. A 3D-printed opaque and portable phone clamp is utilized to hold the smartphone and isolate external light sources, which fluctuate in different environments. Inside the phone clamp, one LED diode is placed which is being used as the light source instead of using the smartphone’s flashlight to get controlled exposure.

### 2.3. Colorimetric Measurement

To quantitatively measure the colorimetric values, ImageJ is utilized. For filter paper measurement, a red-channel value is selected as the primary parameter to reflect the concentration of hemoglobin, because hemoglobin is the compound that contributes red pigment to blood due to the iron atoms it contains.

## 3. Materials and Methods

The fabrication method and materials for all needed parts of our devices are demonstrated is this section; this refers to components from the microfluidic channel and the 3D printed phone clamp.

### 3.1. Microfluidics Channel

The microfluidic channels are assembled by two medical level tapes. The microfluidic channel is engraved by a laser cutter (Boss LS-1420, MinPower (%)-1: 14, MaxPower (%)-1: 14, Speed (mm/s): 9) on double-sided adhesive tape (3M 9965 Diagnostic Microfluidics Tape, 3M, Maplewood, MN, USA). The filter paper is placed in the channel with a hydrophilic cover (3M 9962 Diagnostic Microfluidics Tape, 3M). In Figure 2a, the end of the channels is shown to be the inlet of the channels, and the height of the channel is equal to the height of the double-sided adhesive tapes.

### 3.2. Micropump

A micropump is a powerful active device to accelerate the absorption of the filter paper by accelerating the flow through the channels. To fabricate multiple PDMS micropumps with variable dimensions, we 3D-printed molds with diverse parameters. The rough surface of the 3D printed mold presents challenges when curing PDMS. The acrylic lacquer (Rust-Oleum Acrylic Lacquer) was sprayed onto the surface after 3D printing to form a smooth layer onto the surface of the coarse molds, which will otherwise fail the PDMS curing. It was sprayed a total of three times: the surface was baked for 30 min after the first two times, and a 60-min baking step was needed after the third spraying to ensure the solidity of the acrylic. To fabricate a PDMS structure that feasibly supports the application of the micropump, the ratio of the elastomer and curing agent was adjusted from the normal 10:1 to 20:1, which made it soft enough to press with enough strength to support the structure.

The molds and PDMS micropump (height: 1.2 cm, side: 2.5 cm, pump diameter: 2 cm, pump height: 1 cm) is shown in Figure 2b. Fabricated micropumps with different configurations for the dimensions of height and diameter were used to get the best fit for our device. For this purpose, the theoretical volumes of the pumps were calculated and compared it to the actual volume it can introduce to the sensor. For the height, 3 micropumps were fabricated with different heights of 1 cm, 0.7 cm, and 0.4 cm with a diameter of 2 cm as shown in Figure 2c. From the plot, that 0.7 cm was the optimal height for our application. Additionally, micropumps with different diameters of 2 cm, 1.5 cm, and 0.5 cm with a height of 0.7 cm were made, as shown in Figure 2d, and found that 2 cm gave the optimal results. The density of dyed DI (Deionized) water measured by weighing the solution with a known volume. Then, the weight difference before and after micro-pumping were measured to calculate the pump’s capability.

Figure 3a,b illustrate how the diameter will influence the actual volume more than height. To investigate the actual volume of micropumps, three PMDS micropumps were made using same mold for each size. Then, three devices were made by three micropumps to pump the water into the devices. To measure the weights of the whole device before and after pumping. Thus, the actual volume will be the weight difference divided by water density (1 g/cm${}^{3}$). In Figure 3a, the ratio of theoretical and actual volume is about 3:1, which is larger than that for micropumps with different diameters. However, if the height is too short, the structure cannot support the pressing force so there is no actual volume for a 0.4 cm height in Figure 3a.

### 3.3. Assembly and Operation

To utilize our device conveniently, we combined the extension part with the device shown in Figure 4a; the method to make the glass slide extension is the same as that used when making micropumps.

The glass capillary is utilized as the channel for the solution, which, in turn, is poured into microfluidic channels. Figure 4b shows the final biomedical device after assembling. To operate the device, hold it and press the micropump to the bottom with the thumb and move it closer to the fingers or the blood source. Figure 4c shows the device after absorbing human blood (BioIVT Inc., New York, NY, USA). Then, the sucking volume of the blood can be controlled by changing the parameters of the micropumps to avoid redundant or insufficient blood samples.

### 3.4. Phone Clamp

To better observe the color change, a black opaque phone clamp was 3D-printed (Ultimaker S5, Ultimaker, Utrecht, Netherlands) for an iPhone 12, which is small, portable, and can be modified to fit any device flexibly, as shown in Figure 5a. The whole black phone clamp can control the light source, blocking external light noise. The LED diode (Chanzon white 8mm straw hat LED) is implemented inside to provide a consistent light source [34,35,36]. After absorbing the blood, biomedical devices are slipped into the phone clamp through the sliding window, as shown in Figure 5a. Further, Figure 5b shows the actual photos captured for post-processing. To test the sealing capability of the clamp, photos were taken with the same chip under three different illumination environments (total black room, a room with one ceiling light on, a room with two ceiling lights on) with all other variables same (position, temperature, etc.) and three repeated photos were taken under each environment. The result showed that the average RGB values have negligible difference, proving that the opaque can eliminate all other illumination noise. Then, the smartphone camera focus impact on the RGB values was tested. A dot was drew on the center of the phone screen and manually clicked at the dot to focus before taking photos. The analysis shows that the difference in RGB values is negligible with the same chip and under the same outside illumination environment.

### 3.5. Materials

To validate the method described, 3 different sample types were chosen as hemoglobin resources and tested. The 3 hemoglobin resources are lyophilized bovine hemoglobin powder (Sigma-Aldrich Inc., Burlington, MA, USA), frozen beef blood (Martin Inc., Chesapeake, VA, USA), and Human Blood (BioIVT llc., New York, NY, USA). These 3 different samples were tested using a commercially available hemoglobin analyzer (ACCU-Answer, Johor, Malaysia) [37] for reference. We then imaged the same sample using our experimental setup and utilized ImageJ to measure the red-channel value. This process was repeated for varying concentrations of hemoglobin by diluting the sample with 1x Phosphate-Buffered Saline (PBS) and repeated three times for each concentration. This process for was repeated for all 3 samples, i.e., lyophilized bovine hemoglobin powder, frozen beef blood, and human whole blood.

## 4. Results

Titration curves were obtained by plotting the hemoglobin concentration obtained from the analyzer and the Red channel of the images from ImageJ because of Red channel shows better correlationship than Blue, Green and Average RGB Channel. A logarithmic relationship between the hemoglobin concentration and the red channel can be observed which is summarized in the subsequent sections. In the consecutive sections, the logarithmic models governing the relationship between hemoglobin concentration and the red channel of the RGB value for all 3 sample types will be presented. In the following plots, especially in frozen beef blood and whole human blood, the data points at 10 g/dL are showing shift from the trend line because they are near the end of the dynamic range in our experiments and thus is more prone to offset and bias. Additionally, frozen beef blood and whole human blood have less data points than bovine hemoglobin powder which will affect the bias.

### 4.1. Bovine Hemoglobin Powder

For bovine hemoglobin, we dissolved the powder in a 1X PBS solution (hemoglobin from bovine blood, Sigma-Aldrich llc.). The test concentration ranged from 8 to 20 g/dL. As Figure 6 illustrates, the trendline is logarithmic for the concentration and red-channel value, and the function is y = 41.63984 × ln(26.79662 × ln(x)) and R-value is 0.98615.

### 4.2. Frozen Beef Blood

To prove our theory further, in Figure 7, the concentration from 8 to 20 g/dL beef blood is shown to be performed (frozen beef blood, Martin Inc.). The relation between the concentration of beef blood and the red-channel value is logarithmic with the function y = 77.11186 × ln(4.65844 × ln(x)) and R-value is 0.94524.

### 4.3. Human Blood

Ultimately, we tested real human blood with a concentration ranging from 5 to 11 g/dL (BioIVT llc., New York, NY, USA). In Figure 8, the reddish compound can be observed in an increasing trend from 5 to 11 g/dL directly in actual experiment photos. In Figure 9, the red-channel value of human blood is reflecting a logarithmic relation as a function of different concentrations, which is y = 38.73709 × ln(78.57377 × ln(x)) and R-value is 0.96104.

## 5. Conclusions

In this research paper, we explored an innovative type of microfluidic channel based on 3M medical tapes. Combining a PDMS extension and micropump can offer advantages such as user-friendliness, accuracy, and controllable volume of solution deliverable. Our fabrication method is uncomplicated and flexible, and beginners with little to no experience can operate it perfectly. Furthermore, the shape of the channels can be modified quickly and easily. We described the method for testing the capability of micropumps, so we could choose the right micropump. The improvement made to the light environment is a 3D-printed phone clamp with an LED diode, which provides a sufficient and consistent light source for the measurement of hemoglobin concentration. In this paper, the iPhone 12 was used specifically for the designed 3D printed phone clamp to show the feasibility, and more phone models will be tested in the future to show the universality of this methodology. The regression model was plotted to provide the standard for average red-channel values at different concentrations. As a result, we showed the logarithmic relationship for red-channel values as a function of hemoglobin concentrations. Although we demonstrated the capability of our device for measuring hemoglobin concentration, the colorimetric measurement performed using our biomedical device combined with a smartphone can be extended to other reagents and assays such as glucose measurement. Furthermore, combining with smartphone application, this diagnosis system can be fully portable and Point-of-Care.

## Figures and Tables

**Figure 1 sensors-23-00394-f001:**
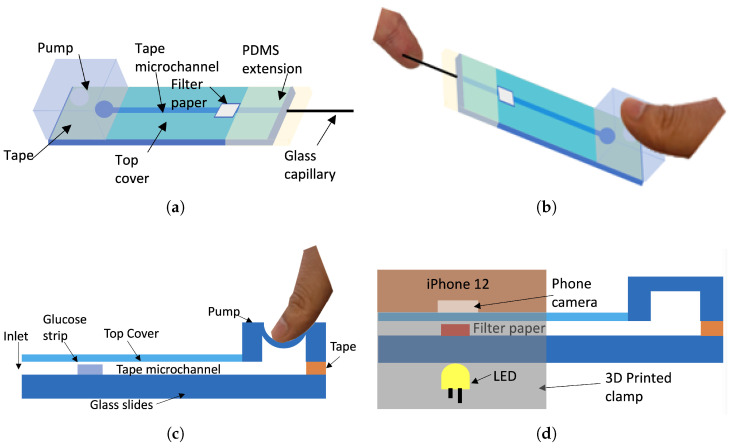
Conceptual pictures of the principle: (**a**) The microfluidic device; (**b**) Dip the microfluidic device in a finger; (**c**) Apply pressure down the micropump to create a negative pressure environment; (**d**) Device in 3D printed phone clamp after adsorbing.

**Figure 2 sensors-23-00394-f002:**
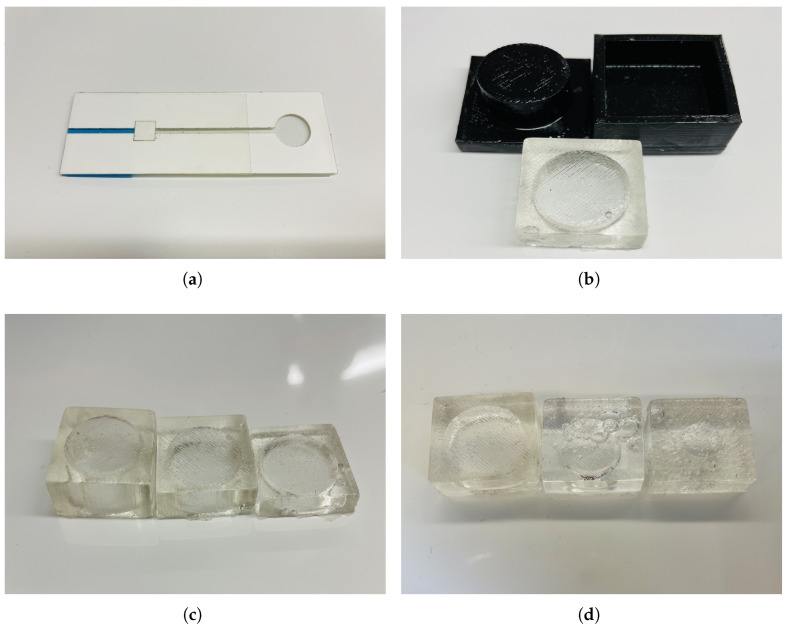
(**a**) Fabricated and assembled microfluidic device; (**b**) 3D-printed PDMS mold with acrylic coating and PDMS micropump; (**c**) Three different PDMS micropumps of 1 cm, 0.7 cm, and 0.4 cm heights with the same 2 cm diameter; (**d**) Three different PDMS micropumps with 2 cm, 1.5 cm, and 0.5 cm diameters with the same 0.7 cm height.

**Figure 3 sensors-23-00394-f003:**
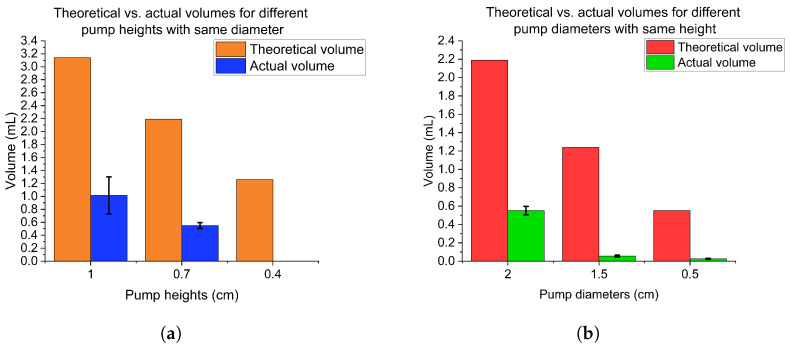
(**a**) Theoretical and actual volumes for three PDMS micropumps with different heights; (**b**) Theoretical and actual volumes for three PDMS micropumps with different diameters.

**Figure 4 sensors-23-00394-f004:**
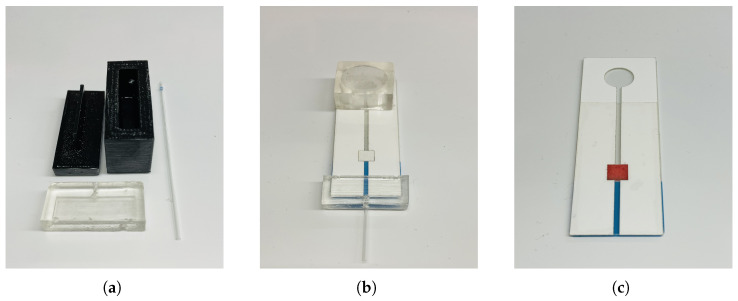
(**a**) 3D-printed mold for the extension part and PDMS extension with dipping glass capillary; (**b**) Whole device after assembling; (**c**) Filter paper with the blood completely absorbed.

**Figure 5 sensors-23-00394-f005:**
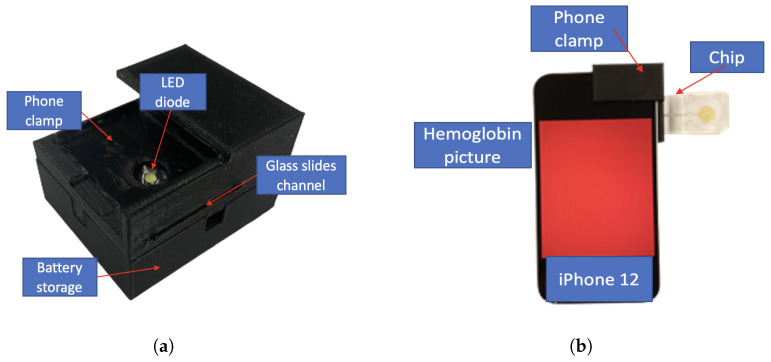
(**a**) 3D-printed portable phone clamp with LED diode mounted; (**b**) Actual photo taken when sliding the glass slide inside the clamp.

**Figure 6 sensors-23-00394-f006:**
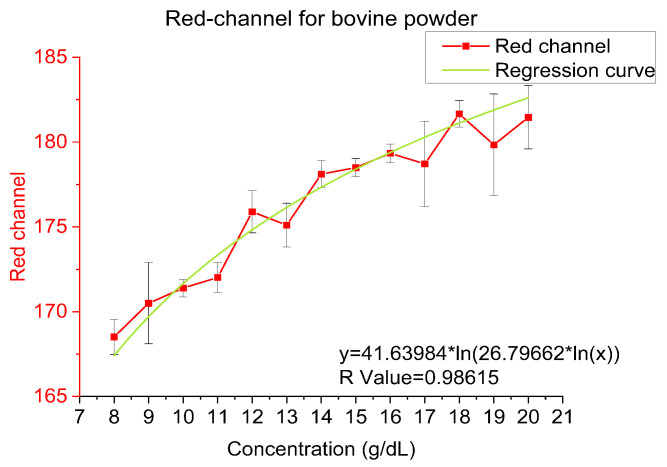
Red-channel value and regression for bovine hemoglobin powder.

**Figure 7 sensors-23-00394-f007:**
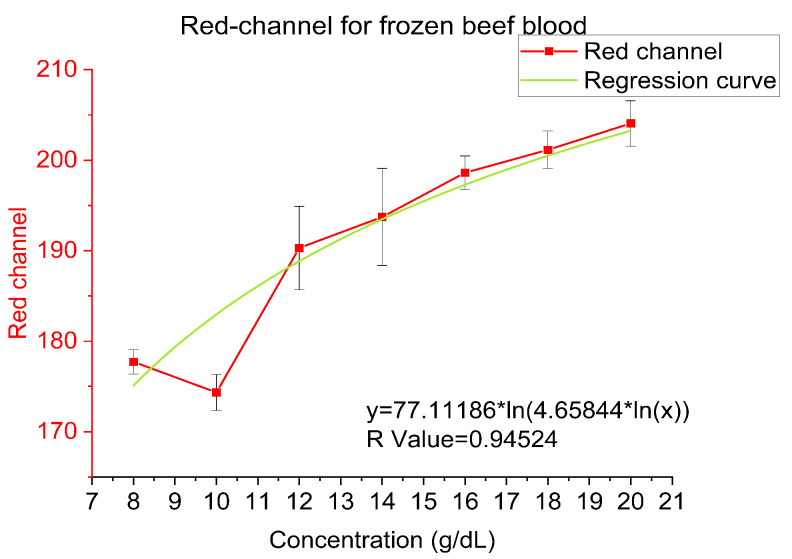
Red-channel value and regression for frozen beef blood.

**Figure 8 sensors-23-00394-f008:**
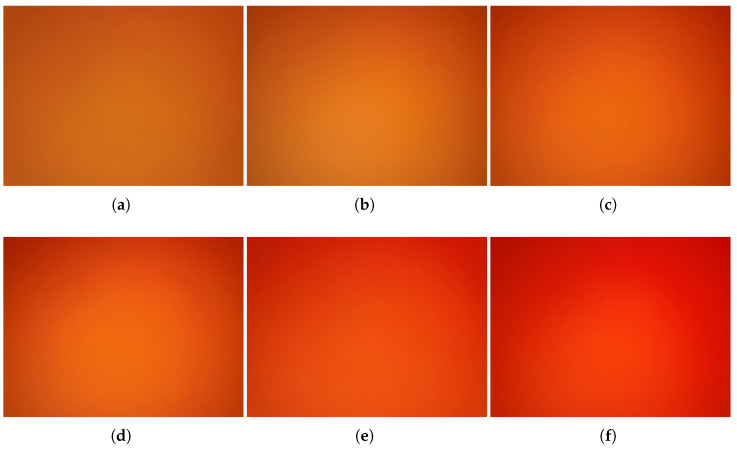
Actual photos of human blood (**a**) 6 g/dL; (**b**) 7 g/dL; (**c**) 8 g/dL; (**d**) 9 g/dL; (**e**) 10 g/dL; (**f**) 11 g/dL.

**Figure 9 sensors-23-00394-f009:**
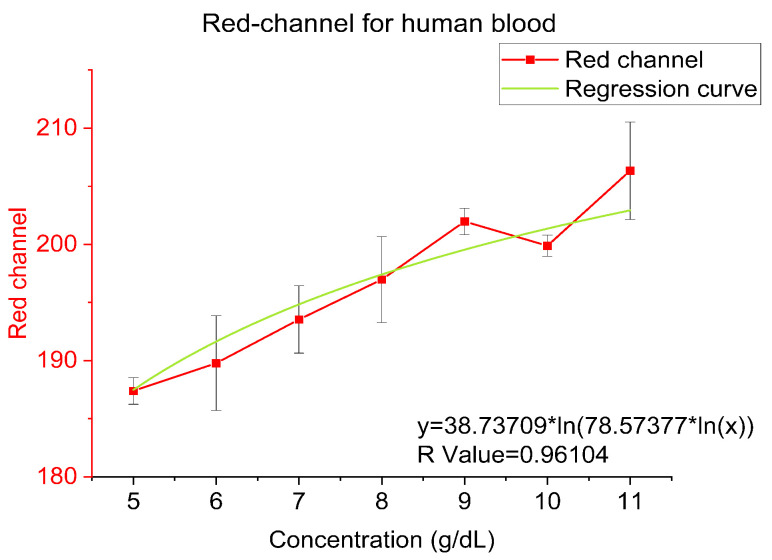
Red-channel value and regression for human blood.

## Data Availability

The pictures dataset taken by instrument and CAD files for phone clamp designs are available from https://github.com/zhuolunmeng/-A-Smartphone-Based-Disposable-Hemoglobin-Biosensor-Based-on-Colorimetric-Analysis (accessed on 1 September 2022).

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
