# Peer review of "A Smartphone-Based Disposable Hemoglobin Sensor Based on Colorimetric Analysis"

_sensors, 2022, doi:10.3390/s23010394_

Round 1
Reviewer 1 Report
The manuscript "sensors-2087871" by Zhuolun Meng et al., describes a protocol for optical determination of hemoglobin levels in whole blood and a complete portable device based on a smartphone camera along with in-house constructed instrumentation to ensure detection stability under sunlight conditions. The device described in the manuscript is interesting and fits well into the current trend of sensors or disposable cassettes development in paper/ adhesive tapes technology for potential application in rapid POCT diagnostics. Undoubtedly such topic is well within the scope of Sensors journal. My comments mainly refer to the poor quality of the graphics and the overall way of graphical presentation of the results. Please find below the comments that I recommend the authors to consider to further improve the manuscript.
1) Line 2: from the point of view of the formal definition of a (bio)sensor as a reusable device, there are no "disposable biosensors." It may be worth considering replacing it with the term "assay" or discussing this aspect in more detail in the body of the manuscript.
2) Line 20: There is a problem with the reference format or an overlooked comment by the Author.
3) Figure 1: The captions of the system elements are far too small and thus unreadable. In addition, it would also be advisable to include a good-quality photograph of the entire system components so that the reader is better oriented with the physical appearance of the system components.
4) The photographs in Figure 2 and Figure 4 are of poor quality - they are all taken with low contrast/light, and Photo 4a is very low resolution. If available, I recommend inserting a higher quality images.
5) In the manuscript we have the plural ("we calculated..."-see line 126) and the singular ("I made...") mixed up. I recommend unification throughout the manuscript.
6) It is worth taking more attention to visual aspects of the presented data. E.g., graph labels and axes in Figure 3 (and others) are very small and densely spaced and thus are illegible. On the other hand, the diagram in Figure 5 is practically completely illegible and the smartphone (Fig. 5b) is not described in any way.
7) Line 166: the sentence is missing a verb.
8) Lines 180-181: „… the Red channel of the images from ImageJ because of Red channel shows better correlationship than Blue, Green and Average RGB Channel” -this sentence needs to be supported by experimental/numerical data or more extensive discussion.
9) Lines 184-185: „We present the linear models governing the relationship between hemoglobin concentration and the red channel of the RGB value for all 3 sample types.” – there is no reference - the reader does not know where the Authors will present these models.
10) It would be useful to provide within the text the information about diagnostically useful ranges of hemoglobin concentrations. Does the dynamic response range of the developed sensor match with the diagnostic range? Please comment more extensively.
11) Fig. 7. For low Hb concentrations (8-19 g/dL) we have an obvious linearity disturbance, worsening the correlation coefficient of the whole model. If this is a coarse error, or there is simply a significant bias, these points should not be considered in model development. Please provide an in-depth explanation of Your decision to use them.
12) Have the authors considered introducing reference measurement into the model? Such as measurement at a different wavelength or measurement of serum samples corresponding to the whole blood samples? Perhaps such a background analysis would help improve the developed model.
13) Authors wrote „…advantages such as user-friendliness, accuracy, and a solution of suitable volume…” – this sentence is logically incorrect (solution cannot be an advantage) - please consider rewriting it.
Reviewer 2 Report
This paper presents an interesting, but as yet no well characterized idea for colorometrically measuring hemoglobin. As of now I don't think enough has been done to characterize and improve the system for it to be published, as evidenced by the very high error in all measurements, the data points that break linearity, and the test having only been performed using 1 model of phone. Additionally I have the following specific comments regarding necessary corrections, and suggestions for improvement:
Line 20 – After reference 1 there is “nathan1966thalassemia” which needs to be removed.
A conceptual picture of the phone clamp should be included in section 2
Section 3.1 – What kind of filter paper was used? Was anything done to the filter paper to make it bind blood cells, or is it just normal paper and the test is reliant entirely on the colour of non-specifically adsorbed blood cells?
Section 3.2 – Includes first person narrative, should be changed to third person
Line 140 – What is this “([per-mode=symbol]1)” ?
Section 3.4 – Also has first person sentences that need to be changed
Were any other phones tested? Multiple different phones should be tested as not all users will own an iPhone 12.
Section 4 contains materials and methodology information that should be in section 3.
Section 4.1 2 and 3 – r values for the lines fit needs to be included
How was the beef blood and human blood hemoglobin diluted? Or did it have a natural variation you measured?
Is Figure 8 labeled correctly? Visually from A to F the pictures appear to be getting more red, but the label states that A is the highest concentration which doesn’t agree with the linear graph.
The error values are very high in your measurements of hemoglobin, limiting the accuracy of this test. How could you improve this?
Can you explain why hemoglobin values of 10 g/dL appear to break the trend for both bovine and human blood?
The trendlines look closer to logarithmic than linear, have you analyzed the data using other fits?
In the conclusions you mention this could be extended to glucose measurement, but you are only measuring the colour of blood. How could such a colorometric test measure the ouput of glucose oxidase?
Reference 5 is incorrectly formatted
Round 2
Reviewer 2 Report
Thank you for the changes made to the paper. I do still think it is lacking especially regarding testing with another model of phones. Although you say that there should be no difference since you are measuring the red channel, you absolutely need to prove this, as manufacturing differences could alter the value. Without this, the test is far too limited.
You also do need to offer up a hypothesis for why the 10 g/dL samples show such a marked shift from the trendline, just saying it's interesting is not enough, you have to offer an explanation or test it again to see if there was an experimental mistake that caused it.
